Influence of the COVID-19 pandemic on breastfeeding support for healthy mothers and the association between compliance with WHO recommendations for breastfeeding support and exclusive breastfeeding in Japan

Nanishi Keiko keiko50@m.u-tokyo.ac.jp 1
Okawa Sumiyo 2
Hongo Hiroko 3
Shibanuma Akira 3
Abe Sarah K. 4
Tabuchi Takahiro 5
1 Office of International Academic Affairs, Graduate School of Medicine, The University of Tokyo , Bunkyo-ku , Tokyo , Japan
2 Institute for Global Health Policy Research, Bureau of International Health Cooperation, National Center for Global Health and Medicine , Shinjuku-ku , Tokyo , Japan
3 Department of Community and Global Health, Graduate School of Medicine, The University of Tokyo , Bunkyo-ku , Tokyo , Japan
4 Division of Prevention, National Cancer Center, Institute for Cancer Control , Chuo-ku , Tokyo , Japan
5 Department of Cancer Epidemiology, Cancer Control Center, Osaka International Cancer Institute , Osaka City , Osaka , Japan
Collado Maria Carmen
Electronic publication date: 2022 May 19
Publication date: 2022
Volume: 10
Electronic Location ID: e13347
Received 2021 Dec 20; Accepted 2022 Apr 6
Copyright: ©2022 Nanishi et al.
Copyright year: 2022
Copyright holder: Nanishi et al.
License: This is an open access article distributed under the terms of the Creative Commons Attribution License, which permits unrestricted use, distribution, reproduction and adaptation in any medium and for any purpose provided that it is properly attributed. For attribution, the original author(s), title, publication source (PeerJ) and either DOI or URL of the article must be cited.
License URL: https://creativecommons.org/licenses/by/4.0/

Keywords: COVID-19, Professional breastfeeding support, Ten steps to successful breastfeeding, Exclusive breastfeeding, Japan

Funding: Japan Society for the Promotion of Science KAKENHI 18H03062 21H04856 This study was funded by the Japan Society for the Promotion of Science (JSPS) KAKENHI Grants, 18H03062 (Takahiro Tabuchi) and 21H04856 (Takahiro Tabuchi and Sumiyo Okawa). The funders had no role in study design, data collection and analysis, decision to publish, or preparation of the manuscript.

==============================
Background

Professional breastfeeding support contributes to maternal and child health. However, the influence of the current coronavirus disease 2019 (COVID-19) pandemic on breastfeeding support has not been carefully examined. Therefore, we assessed maternal breastfeeding intention and professional breastfeeding support before and during the pandemic. We further examined the association of compliance with World Health Organization (WHO) recommendations for professional breastfeeding support with exclusive breastfeeding during the pandemic.

Methods

This cross-sectional, internet-based, questionnaire study analyzed data from 484 healthy women with live singleton births between 15 October 2019 and 25 October 2020 in Japan. A delivery before 5 March 2020 was classified as a before-pandemic delivery (n = 135), and a delivery after 6 March 2020 was a during-pandemic delivery (n = 349). Among the ten breastfeeding support steps recommended by the WHO, we assessed the five steps that are measurable by maternal self-report and would likely exhibit variability. Receipt of a free formula sample or invitation to a free sample campaign by the time of survey was also asked. Infant feeding status at the time of the survey was measured among women with infants younger than 5 months, which was a subgroup of mothers who delivered during the pandemic. Mothers were asked what was given to infants during the 24 h before the survey and when nothing other than breast milk was given, the status was classified as exclusive breastfeeding.

Results

While 82.2% of women with a delivery before the pandemic intended to breastfeed, the rate was 75.6% during the pandemic (p = 0.120). The average number of breastfeeding support steps received was 3.24 before the pandemic but it was 3.01 during the pandemic (p = 0.069). In particular, rooming-in was less frequent (39.3% before vs. 27.8% during the pandemic, p = 0.014). Among mothers with infants younger than 5 months who had a delivery during the pandemic (n = 189), only 37.0% (n = 70) reported exclusively breastfeeding during the 24 h before completing the survey. Multiple logistic regression analysis indicated that receiving support for all five steps was positively associated with exclusive breastfeeding during the 24 h before the survey (adjusted odds ratio 4.51; 95% CI [1.50–13.61]). Receipt of a free formula sample or invitation to a free sample campaign was negatively associated with exclusive breastfeeding (adjusted odds ratio 0.43; 95% CI [0.19–0.98]). Other factors related to non-exclusive breastfeeding were older maternal age, lower education level, primiparity, and no breastfeeding intention.

Conclusions

The pandemic weakened breastfeeding support for healthy women in Japan; however, support practice that adhered to WHO recommendations appeared to be effective during the pandemic.

Introduction

The COVID-19 pandemic has changed people’s lives. A key strategy in controlling the pandemic is maintaining physical distance from other people. Some countries implemented urban lockdowns, and many restricted social activities in order to reduce contact between people. In Japan, although no city lockdown has been ordered, during periods of high COVID-19 transmission the national and local governments have requested people to stay home, work remotely, and avoid inter-prefectural travel and urged restrictions on entry to shopping malls and other places that encourage people to leave their homes. Although attention is understandably focused on patients with severe COVID-19, uninfected persons are affected by isolation from social networks that help them choose healthy behaviors and maintain health and well-being. Pregnant women and mothers with infants require particular forms of assistance from professionals, friends, and family, as they must maintain their health and that of their infants, adopt child care skills, initiate and continue breastfeeding, and combine childrearing with their other responsibilities (McFadden et al., 2017; Negron et al., 2013). Such women are thus among those most affected by the pandemic (Kotlar et al., 2021).

Breastfeeding benefits maternal and child health (Rollins et al., 2016; Victora et al., 2016), and because it is a learned behavior, women need continuous support during pregnancy, at delivery, and after birth (McFadden et al., 2017; Brown, 2017; Emmott, Page & Myers, 2020; Nanishi, Green & Hongo, 2021), even during the COVID-19 pandemic. However, physical distancing and an overextended health system could reduce appropriate support and discourage women from starting and continuing breastfeeding (Vazquez-Vazquez et al., 2021; Ceulemans et al., 2020; Brown & Shenker, 2021). A study of UK mothers who had breastfed their babies aged 0–12 months at least once during the COVID-19 pandemic found that some reported that the pandemic positively influenced breastfeeding because of the increased time at home, reduced pressure, and presence of fewer visitors. For those who lacked sufficient access to breastfeeding support, however, breastfeeding during the pandemic was challenging (Brown & Shenker, 2021). Mothers with less education and more challenging living circumstances, and those of Black or other minority ethnic backgrounds, were more likely to stop breastfeeding during the pandemic (Brown & Shenker, 2021).

Health professionals are crucial to successful breastfeeding, but the pandemic could indirectly affect breastfeeding support. The World Health Organization (WHO) recommends ten steps to support breastfeeding (World Health Organization, 2017; World Health Organization, UNICEF, 2009), and existing evidence indicates that compliance with these recommendations improves breastfeeding outcomes (Hannula, Kaunonen & Tarkka, 2008; Merten, Dratva & Ackermann-Liebrich, 2005). Specifically, mothers are more likely to breastfeed when professionals help them practice skin-to-skin contact and initiate breastfeeding soon after birth, learn sufficient breastfeeding skills, stay roomed-in with their infant, and respond to infants’ cues (Merten, Dratva & Ackermann-Liebrich, 2005). However, health professionals may deprioritize such practices during a pandemic, when they lack a clear message encouraging breastfeeding support.

The indirect effects of the COVID-19 pandemic on professional breastfeeding support have not been carefully examined. This is of particular interest because women become mothers and need continuous support for breastfeeding even during pandemics, and breastfeeding promotes infant and maternal health. A study in Italy suggested that the rate of exclusive breastfeeding was lower during than before the pandemic (Latorre et al., 2021). However, the extent to which professional breastfeeding support is affected by the pandemic is unknown. Therefore, we examined maternal breastfeeding intention and professional breastfeeding support before and during the pandemic in Japan and analyzed any differences. We further assessed if compliance with WHO recommendations for infant feeding support was effective in promoting exclusive breastfeeding during the pandemic.

Materials & Methods

Study design and participants

We used the STROBE cross sectional checklist when writing our report (von Elm et al., 2007) (Table S1). This cross-sectional, internet-based questionnaire survey was conducted as a part of the Japan COVID-19 and Society Internet Survey (JACSIS) study, which addresses public health issues related to the COVID-19 pandemic. The study sample for the project was retrieved from the pooled panels of an internet research agency (Rakuten Insight, 2016), which had approximately 2.2 million panelists in Japan in 2019. The current study targeted pregnant and postpartum women. Among the 21,896 women in the panel who had given birth after 1 October 2019 or were expected to give birth by 31 March 2021, 4,373 were selected by simple random sampling and invited to the survey by e-mail. Data collection was started on 15 October 2020 and ended on 25 October 2020, when the targeted sample size of 1,000 was attained (response rate, 22.9%). The sample size of 1,000 was decided based on feasibility to conduct the research. Details of the sampling strategy are described elsewhere (Okawa et al., 2022).

The inclusion criteria of the study sample were women who had a live birth from 1 October 2019 through 25 October 2020 and pregnant women expected to have a delivery by 31 March 2021. Out of the 1,000 pregnant and postpartum women who participated, we excluded those with any invalid response (n = 74) and those still pregnant at the time of survey (n = 400), because the focus of this study was breastfeeding support and practice. Among the 558 women who had a live birth by the time of the survey and gave valid responses, those with a medical condition that could affect breastfeeding were excluded. Specifically, we excluded women who gave birth before 37 weeks of gestation (n = 23), those whose infants were admitted to a neonatal intensive care unit (n = 39), those with a health condition that stopped them from seeing their infant for longer than 1 day (n = 35), those who had multiple births (n = 14), and those who had undergone cesarean delivery because of COVID-19 (n = 1). Some mothers satisfied multiple exclusion criteria. Ultimately, 484 women with a healthy live singleton birth were included in the analysis (Fig. 1).

Figure 1 Participants flow diagram.

This is a flow diagram of participants,.where “n” stands for number of women who approached and/or participated to the Japan COVID-19 and Society Internet Survey (JACSIS) study, which was conducted online between 15 October 2020 and 25 October 2020. Those with a medical condition that could affect breastfeeding were excluded from the analysis and some mothers satisfied multiple conditions.

A delivery on 5 March 2020 or earlier was classified as a before-pandemic delivery and a delivery on 6 March 2020 or later as a during-pandemic delivery. The precise start date of the pandemic is a matter of debate. The first case of COVID-19 in Japan was detected on 14 January 2020, and the first state of emergency, declared on 7 April 2020, originally targeted seven of the 47 prefectures and was later expanded to include the entire country. This pattern reflects how COVID-19 cases per population differed by prefecture. By 12 March 2022, Japan had 5.7 million diagnosed cases and 26 thousand deaths of COVID-19 (Johns Hopkins University and Medicine CRC , 2022). The deaths per 100,000 population were 20.65 on 13 March 2022, which was below many OECD countries, such as 294.77 in the United States, and 151.06 in Germany (The Johns Hopkins Coronavirus Resource Center, 2022). Because the focus of this study was the indirect effect of the pandemic on breastfeeding, we decided to establish a pandemic start date that reflected the time point when health professionals’ infant feeding care might have changed. The three major obstetrics academic societies in Japan published a joint statement on 5 March 2020, which recommended that if a woman with COVID-19 has a delivery, the infant should be completely separated from the mother and breastfeeding avoided (Japan Society for Infectious Diseases in Obstetrics and Gynecology, Japan Society of Obstetrics and Gynecology & Japan Association of Obstetricians and Gynecologists, 2021). Because this statement could have changed professional practices, we defined the cut-off date as 5 March 2020. Two major academic societies, in pediatrics and neonatology, later stated that breast milk was less likely to transmit SARS-CoV-2 and thus included breastfeeding as a feeding option even when a mother had received a diagnosis of COVID-19 (Japan Pediatric Society, 2021; Japan Society for Neonatal Health and Development, 2021). Nevertheless, the policy of mother–infant separation has not changed as of this writing. The nationwide survey investigated 540 women diagnosed with COVID-19 during pregnancy between 1 January 2020 and 31 January 2022, and found that 94% were separated from their newborns and 59% formula-fed entirely at a hospital if they delivered within two weeks of the diagnosis (Deguchi & Yamada, 0000).

Measurements

Breastfeeding support from medical professionals

Professional breastfeeding support was measured in two ways: by assessing infant feeding care provided by obstetric ward staff and by examining uptake of regular maternal and child health services from pregnancy through the postpartum period. Infant feeding support by obstetric ward staff was measured in relation to compliance with the Ten Steps to Successful Breastfeeding (hereafter referred to as the Ten Steps) recommended by the WHO and Unicef (World Health Organization, UNICEF, 2009). Among the ten steps recommended, we measured five steps that could be assessed by means of maternal self-report and would likely exhibit variability among mothers in Japan. Table S2 summarizes the ten steps and the basis of the selection of the five steps. Briefly, we measured receipt of five steps at the hospital, namely, Steps 3, 4, 5, 7, and 8, which correspond to “discuss the importance and management of breastfeeding with women and their families,” “facilitate immediate skin-to-skin contact and support mothers to initiate breastfeeding as soon as possible after birth,” “support mothers to manage common breastfeeding difficulties,” “enable mothers and their infants to practice rooming-in 24 h a day,” and “support mothers to respond to their infants’ cues for breastfeeding every time.” These questions were developed by the authors. We carefully chose plain Japanese language for each question (Table S3) and did not conduct a pilot test among mothers.

In Japan, antenatal and postpartum health services for the mother, parents, and infants are an opportunity to augment breastfeeding counseling provided as part of in-hospital care. To measure access to professional breastfeeding support during this routine care, participants were asked to recall provision of antenatal health check-ups and mother/parent classes, newborn home visits, and infant health check-ups and immunizations.

Infant feeding status

Infant feeding status at the time of the survey was measured among women with infants younger than 5 months. The WHO and many other organizations recommend exclusive breastfeeding during the first 6 months of life (World Health Organization, 2002; Section on Breastfeeding, American Academy of Pediatrics, 2012). One of the indicators to monitor the achievement of the goal is exclusive breastfeeding under six months; the percentage of infants 0–5 months of age who were fed exclusively with breast milk during the previous day. Specifically, the numerator is infants 0–5 months of age who were fed only breast milk during the previous day, and the denominator is infants 0–5 months of age (World Health Organization, UNICEF, 2021). This point-in-time “exclusive breastfeeding” rate is often used in cross-sectional studies (Gupta et al., 2017) in order to include infants under 6 months and obtain a large enough sample size with reasonable cost (Greiner, 2014). Because the current study is a cross-sectional study, we adopted the point-in-time measurement to assess the infant feeding status. There is no recommendation for the duration of exclusive breastfeeding in Japan. Instead, the Japanese guidelines for professional support of infant and young child feeding recommend starting complementary feeding at age 5 to 6 months (Study Group on Revision of Support Guide for Breastfeeding and Weaning, 2019). Therefore, the breastfeeding status of infants younger than 5 months was measured in this study. Mothers were asked what was given to infants during the 24 h before the survey. Infant feeding status was then classified as exclusive breastfeeding (i.e., no foods or liquids given other than breast milk), combination feeding breastfeeding (i.e., breast milk and formula milk or other foods/liquids was given), and formula feeding (i.e., no breast milk was given). It should be noted that “exclusive breastfeeding” measured in the current study corresponds to the indictor of infant feeding status recommended by the WHO and the United Nations Children’s Fund (UNICEF) (World Health Organization, UNICEF, 2021) and does not indicate exclusive breastfeeding from birth to six months of age, which is a public health goal recommended by the WHO (World Health Organization, UNICEF, 2009).

Breastfeeding intention and other factors that might affect breastfeeding

In addition to maternal intention of breastfeeding, the following factors known to be associated with breastfeeding were analyzed: socioeconomic background, the mental condition of mothers, and social factors, including work status and marketing of breast milk substitutes. Mothers described their original infant feeding intention as “definitely wanted to breastfeed,” “wanted to breastfeed, if possible,” “wanted to feed a combination of breast milk and formula milk,” and “did not have a specific plan.” Those who chose the first two options were considered to have an intention to breastfeed.

Analysis

The chi-square test or Fisher’s exact test was used to compare mothers who delivered before and during the pandemic in relation to their characteristics, breastfeeding intention, and provision of breastfeeding support. Descriptive statistics are used to describe the infant feeding status of mothers with an infant younger than 5 months. Finally, logistic regression analysis was used to examine if provision of infant feeding support that adhered to the Ten Steps was associated with exclusive breastfeeding of infants younger than 5 months during the previous day of the survey. Factors known to be associated with breastfeeding were considered potential confounders and entered into the model. On the basis of previous studies, the potential confounders evaluated were age, marital status, education level, household income, work status, mode of delivery, parity, depressive symptoms, intention, support from family and the partner, marketing of breast milk substitutes, and professional support through regular health check-ups (i.e., mother/parent classes and newborn home visits in this study) (Kaneko et al., 2006; Donath, Amir & Team, 2003; Amir & Donath, 2008; Cohen et al., 2018; Dennis & McQueen, 2009). However, marital status, work status, and uptake of newborn home visits were not included because of their lack of variability. Analyses were restricted to individuals with complete data for all variables required for each analysis. A p-value of 0.05 or less was considered to indicate statistical significance. All analyses were conducted using SPSS version 27.0.

Ethical considerations

This study was approved by the Institutional Review Board of the Osaka International Cancer Institute (No. 20084) and the Ethical Committee of the Graduate School of Medicine, the University of Tokyo (No. 2020336NI). This internet survey was conducted anonymously. Not answering the questionnaire was deemed as non-consent.

Results

Characteristics of participants

Table 1 shows the background characteristics of the participants, including infant feeding intention. Mothers who delivered before the pandemic (n = 135) and during the pandemic (n = 349) were compared. In addition, the characteristics of mothers with an infant younger than 5 months (i.e., those who had reported infant feeding methods at the time of the survey) are presented. These 189 mothers are a subgroup of the 349 mothers who delivered during the pandemic. The two groups did not significantly differ in relation to socioeconomic background or cesarean delivery rate, but intention to breastfeed during the pandemic was lower for mothers who delivered during the pandemic than for those who delivered before the pandemic. While 82.2% of women with a delivery before the pandemic intended to breastfeed, the rate was 75.6% during the pandemic (p = 0.120). During the pandemic, only 33.2% of mothers had a delivery in the presence of their partner, and more than 95.1% of mothers reported that family members were not allowed to visit the hospital. Support from family and friends also changed during the pandemic. As compared with mothers who delivered before the pandemic, those who delivered during the pandemic were more likely to report that they could not receive child care support from the infant’s grandparents (14.1% before the pandemic vs. 27.5% during the pandemic; p = 0.002) . The mean age of the infants in the subgroup of mothers with an infant younger than 5 months was 73.9 days (Standard Deviation 49.8).

Table 1 Characteristics of participants, breastfeeding intention, exposure to marketing of breast milk substitutes, and delivery and child-rearing environment.

The second column from the right is the data of a subgroup of those who delivered during the pandemic and also had an infant under five months of age. The rightmost column compares those who gave birth before the pandemic and those who gave birth during the pandemic.

	Delivery before
pandemica
n = 135	Delivery during pandemicb
n = 349	Infant age <5 months at time of survey c
n = 189	Delivery before vs. during pandemic
p value d	
Age (Mean, Standard Deviation)	32.1 (4.1)	32.3 (3.4)	32.2 (4.1)	0.726	
Less than 16 years of formal education	64 (47.4%)	175 (50.4%)	96 (51.3%)	0.551	
Single mother	0 (0 %)	3 (0.9%)	0 (0 %)	0.280 e	
Unintended pregnancy	22 (16.3%)	50 (14.3%)	25 (13.2%)	0.585	
Low household incomef	27 (20.0%)	55 (15.8%)	27 (14.3%)	0.265	
Cesarean delivery	28 (20.7%)	60 (17.2%)	28 (14.8%)	0.364	
Partner present during labor	80 (59.3%)	116 (33.2%)	59 (31.2%)	<0.001	
Family members not allowed to visit after delivery	21 (15.6%)	332 (95.1%)	181 (95.8%)	<0.001	
Primipara	70 (51.9%)	178 (51.0%)	85 (45.0%)	0.867	
Intended breastfeeding during pregnancy	111 (82.2%)	264 (75.6%)	143 (75.7%)	0.120	
Ever received free infant formula sample or invitation to free infant formula campaign	102 (75.6%)	267 (76.5%)	152 (80.4%)	0.826	
Unable to receive childcare support from grandparents after delivery	19 (14.1%)	96 (27.5%)	45 (23.8%)	0.002	
Unable to talk with friends for infant feeding and caring advice	14 (10.4%)	53 (15.2%)	24 (12.7%)	0.169	
Currently working outside home	22 (16.3%)	18 (5.2%)	8 (4.2%)	<0.001	
Edinburgh Postnatal Depression Scale
scoreg (Mean, Standard Deviation)	8.2 (5.7)	6.7 (4.9)	6.7 (4.9)	0.015	
Partner support scale scoreh (Mean, Standard Deviation)	6.9 (2.4)	7.4 (2.1)	7.5 (2.0)	0.021	
Notes.

a Delivery on 5 March 2020 or earlier in Japan.

b Delivery on 6 March 2020 or later in Japan.

c Subgroup of mothers who delivered during the pandemic.

d The chi-square test or T-test was performed, if not specified.

e Fisher’s exact test was performed.

f Annual household income of ≤4 million yen (about 36,000 USD).

g A higher score indicates more depressive symptoms at the time of the survey, not status at delivery.

h Measured with three questions with a four-point Likert scale developed by the authors. The total score ranges from 0 to 9. A higher score indicates more support for childcare from the partner at the time of the survey, not status at delivery. The Cronbach’s alpha for the scale in this study was 0.90.

Infant feeding support from medical professionals before and during the pandemic

Table 2 compares infant feeding support from medical professionals for mothers with deliveries before and during the pandemic. Women who delivered during the pandemic were more likely to report no or reduced opportunity to attend classes for mothers/parents (33.3% before the pandemic vs. 75.1% during the pandemic; p < 0.001). However, most did not reduce the frequency of antenatal care visits and did not opt-out of the antenatal home visit. There was no significant difference in receipt of such care between the two groups (p = 0.150 and p = 0.084, respectively). Only 5.4% skipped a scheduled infant health check-up or immunization during the pandemic.

Table 2 Infant feeding support from medical professionals.

The second column from the right is the data of a subgroup of those who delivered during the pandemic and also had an infant under five months of age. The rightmost column compares those who gave birth before the pandemic and those who gave birth during the pandemic.

	Delivery before
pandemica
n = 135	Delivery during
pandemicb
n = 349	Infant age
<5 months at
time of survey
n = 189	Delivery before
vs. during pandemic
(p value c)	
Reduced frequency of antenatal care visits	3 (2.2%)	19 (5.4%)	11 (5.8%)	0.150d	
No or reduced chance to attend maternal/parent classes during pregnancy	45 (33.3%)	262 (75.1%)	155 (82.0%)	<0.001	
No postnatal home visite	15 (11.1%)	61 (17.5%)	19 (10.1%)	0.084	
Skipped infant health check-up or immunizatione	18 (13.3%)	19 (5.4%)	7 (3.7%)	0.003f	
Provision of the Ten Steps to Successful Breastfeeding					
Step 3g	115(85.2%)	274 (78.5%)	149 (78.8%)	0.097	
Step 4h	62 (45.9%)	139 (39.8%)	76 (40.2%)	0.222	
Step 5i	106 (78.5%)	275 (78.7%)	146 (77.2%)	0.947	
Step 7j	53 (39.3%)	97 (27.8%)	50 (26.5%)	0.014	
Step 8k	101 (74.8%)	264 (75.6%)	143 (75.7%)	0.849	
Number of steps practiced (Mean, Standard Deviation)	3.24 (1.24)	3.01 (1.26)	2.98 (1.25)	0.069	
Received all five steps	23 (17.0%)	43 (12.3%)	22 (11.6%)	0.175	
Notes.

a Delivery on 5 March 2020 or earlier in Japan.

b Delivery on 6 March 2020 or later in Japan.

c Comparison between mothers who had a delivery before and during the pandemic.

d Fisher’s exact test was performed.

e Includes both cancellation from the service provider and opt-out by the mother.

f Confounded by infant age; infants born before the delivery had more opportunity to be invited to infant health check-ups and immunization by the time of the survey.

g Learned during the pregnancy the benefits and techniques of breastfeeding.

h Initiated breastfeeding within 30 minutes of delivery.

i Learned in hospital how to feed the baby in the manner desired.

j Roomed-in with the baby since soon after birth.

k On-demand feeding (breastfed every time the baby demanded) in hospital.

Mothers who delivered during the pandemic were less likely to receive breastfeeding support that complied with WHO recommendations than were those who delivered before the pandemic. Analysis of the five steps showed that significantly fewer women received Step 7 (roomed-in with their baby since soon after birth) during the pandemic (39.3% before the pandemic vs. 27.8% during the pandemic; p = 0.014). In addition, fewer women satisfied Step 3 (learned enough during pregnancy about the breastfeeding benefits and techniques; 85.2% before the pandemic vs. 78.5% during the pandemic; p = 0.097) and Step 4 (initiation of breastfeeding within 30 min of delivery; 45.9% before the pandemic vs. 39.8% during the pandemic; p = 0.222), although the differences were not significant. The average number of steps achieved in compliance with the Ten Steps was 3.24 before the pandemic and 3.01 during the pandemic (p = 0.069). Only 17% of those who delivered before the pandemic reported the provision on all five steps measured, and the rate was 12.3% during the pandemic (p = 0.175).

Feeding status of infants younger than 5 months

Infant feeding status at the time of survey was analyzed among 189 mothers who delivered during the pandemic and had infants younger than 5 months; only 70 (37.0%) reported that they exclusively breastfed during the 24 h before the survey (Table 3). Even among the 143 mothers who intended exclusive breastfeeding during pregnancy, only 61 (42.7%) reported exclusive breastfeeding during the 24 h before the survey.

Table 3 Feeding status of infants younger than 5 monthsa.

The second left column shows data of all mothers with infants younger than five months. The second right and rightmost columns showed data by original feeding intention during pregnancy.

	All mothers with infants younger than 5 months
(n = 189)	Original feeding intention during pregnancy	
		Breastfeeding
(n = 143)	Mixed feeding, formula feeding, or no clear intention (n = 46)	
Exclusive breastfeeding b	70 (37.0%)	61 (42.7%)	9 (19.6%)	
Partial breastfeeding b	96 (50.8%)	67 (46.9%)	29 (63.0%)	
Formula feeding b	23 (12.2%)	15 (10.5%)	8 (17.4%)	
Notes.

a Subgroup of participants who had a delivery during the pandemic (on 6 March 2020 or later) in Japan.

b Measured by a 24-hour recall.

Number of steps practiced and feeding status of infants younger than 5 months

Table 4 shows the feeding status of infants younger than 5 months, stratified by the number of the breastfeeding support steps received. Among mothers reporting that they received 0 to 4 steps, the rate of exclusive breastfeeding during the 24 h before the survey ranged from 27.8% to 40.0%. However, among the 22 mothers who reported receipt of all five steps, the rate was 72.7%.

Table 4 Feeding status of infants younger than 5 monthsa, by number of breastfeeding support steps received.

The leftmost column shows the number of steps practiced and the number of the mothers who received that number of practice. For example, five mothers received no step. Each data in the other columns show the numbers of mothers/total mothers responded (%) and the number of mothers by infant feeding status.

Number of steps practicedb (n = 189)	Exclusive breastfeeding	Partial breastfeeding	Formula feeding	
0 (n = 5)	40.0% (2)	20.0% (1)	40.0% (2)	
1 (n = 18)	27.8% (5)	55.6% (10)	16.7% (3)	
2 (n = 42)	31.0% (13)	54.8% (23)	14.3% (6)	
3 (n = 56)	32.1% (18)	55.4% (31)	12.5% (7)	
4 (n = 46)	34.8% (16)	56.5% (26)	8.7% (4)	
5 (n = 22)	72.7% (16)	22.7% (5)	4.5% (1)	
Notes.

a Measured by a 24-hour recall among the subgroup of participants who had a delivery during the pandemic (on 6 March 2020 or later) in Japan.

b Among the Ten Steps to Successful Breastfeeding recommended by the WHO, provision of five steps (Steps 3, 4, 5, 7, and 8) were assessed.

Factors associated with exclusive breastfeeding during the 24 h before the survey

Multiple logistic regression analysis indicated that, after adjusting for possible confounders, provision of all five steps of the Ten Steps promoted exclusive breastfeeding during the 24 h before the survey of infants younger than 5 months (adjusted odds ratio 4.51; 95% CI [1.50–13.61]). In addition, receipt of a free formula sample or invitation to a free sample campaign was significantly associated with a reduction in exclusive breastfeeding during the 24 h before the survey (adjusted odds ratio 0.43; 95% CI [0.19–0.98]), as were older maternal age, lower education level, primiparity, and breastfeeding intention (Table 5).

Table 5 Factors associated with exclusive breastfeeding of infants younger than 5 monthsa.

Each data indicates an unstandardized coefficient, p-value, adjusted odds ratio, and its lower and upper limits of the 95% confidence interval of each variable entered in the model.

	Unstandardized coefficient	P	Adjusted Odds Ratio	Lower limit of the 95% CI	Upper limit of the 95% CI	
Age	−0.122	0.010	0.885	0.807	0.971	
More than university graduate level of education	0.856	0.020	2.354	1.146	4.836	
Low household income	0.381	0.516	1.464	0.463	4.630	
Cesarean delivery	−0.458	0.401	0.633	0.217	1.842	
Primiparity	−1.073	0.008	0.342	0.154	0.759	
Edinburgh Postnatal Depression Scale score	−0.021	0.594	0.98	0.908	1.057	
Intended breastfeeding during pregnancy	0.936	0.043	2.551	1.031	6.31	
Partner support scale scoreb	−0.041	0.662	0.960	0.799	1.153	
Unable to receive childcare support from grandparents after delivery	−0.377	0.395	0.686	0.288	1.635	
Ever received free infant formula sample or invitation to a free infant formula campaign	−0.852	0.046	0.427	0.185	0.984	
No or reduced opportunity for parental/maternal class during pregnancy	0.682	0.161	1.978	0.761	5.136	
Received all five steps	1.507	0.007	4.514	1.497	13.61	
Notes.

a Measured by a 24-hour recall among the subgroup of participants who had a delivery during the pandemic (on 6 March 2020 or later) in Japan.

b Measured using three questions with a four-point Likert scale developed by the authors. The total score ranges from 0 to 9. A higher score indicates more support for childcare from the partner at the time of the survey, not status at delivery. The Cronbach’s alpha for the scale in this study was 0.90.

Discussion

This study of healthy mothers in Japan found that mothers with a delivery during the COVID-19 pandemic received less breastfeeding support than mothers who had a delivery before the pandemic. Compared with mothers who delivered before the pandemic, those who delivered during the pandemic were less likely to receive professional breastfeeding support that complied with WHO recommendations. Although the difference was not significant, mothers who delivered during the pandemic were less likely to intend to breastfeed than were those who delivered before the pandemic. Moreover, among those who had an infant younger than 5 months, there was a wide gap between their original breastfeeding intention and their practice at the time of the survey. These findings suggest that the COVID-19 pandemic affected breastfeeding intention, professional breastfeeding support, and breastfeeding outcomes of healthy mothers.

We found that the quality of in-hospital infant feeding care might be affected by the pandemic, even when accessibility to maternal and infant health services might not be seriously disrupted. Previous studies in low- and middle-income countries suggested that by decreasing access to health care, the COVID-19 pandemic might have considerable indirect effects on maternal and child health (Roberton et al., 2020). In contrast, this study found no significant reduction in the uptake of antenatal care, neonatal home visits, or infant health check-ups. However, compliance with WHO recommendations on breastfeeding support was reduced. In particular, the rate of mothers who achieved Step 7 (“roomed-in with the baby since soon after birth”) was lower during the pandemic. Because mothers with a medical condition or with an infant who had a medical condition were excluded from the analysis, this finding suggests that more mothers were separated from their infants for non-medical reasons during the pandemic. Mother–infant separation may affect maternal and infant well-being (Bartick et al., 2021), increases the risks of breastfeeding difficulties (Cohen et al., 2018; Moore et al., 2016), and interrupts mother–infant interactions (Dumas et al., 2013; Bystrova et al., 2009). To reduce the indirect negative impact of the COVID-19 pandemic on maternal and child health, clear messaging and guidance should be provided to health professionals, to ensure they are able to maintain the highest possible quality of infant feeding care.

During the pandemic, receipt of WHO-compliant breastfeeding support was positively associated with exclusive breastfeeding of infants younger than 5 months, which was measured by 24-hour recall. Most of our participants, however, did not receive such support. This association is consistent with multiple pre-pandemic studies that provided evidence supporting the Ten Steps (World Health Organization, 2017). We evaluated receipt of five steps out of the Ten Steps and found that partial receipt of the five steps does not ensure exclusive breastfeeding. Several possibilities could explain why providing all five steps is necessary for effective support. The first is that “full support,” namely, receipt of Steps 3, 4, 5, 7, and 8, encourages exclusive breastfeeding. Full support encompasses informative support during pregnancy (i.e., Step 3: “discuss the importance and management of breastfeeding with women and their families”), physical contact between the infant and mother soon after birth (i.e., Step 4: “facilitate immediate skin-to-skin contact and support mothers to initiate breastfeeding as soon as possible after birth,” and Step 7: “enable mothers and their infants to practice rooming-in 24 h a day”), and practical help on breastfeeding techniques (i.e., Step 5: “support mothers to manage common breastfeeding difficulties,” and Step 8: “support mothers to respond to their infants’ cues for breastfeeding every time”). The absence of any step of full support could hinder breastfeeding. Hospital characteristics may also explain the positive association between full support and exclusive breastfeeding. Hospitals providing full support might have specific characteristics that enabled mothers to breastfeed exclusively. For example, some of the mothers who reported receiving all five steps might have delivered at Baby Friendly–certified hospitals, which provide infant feeding practices that are consistent with the Ten Steps. However, because only 2% of obstetric wards are Baby Friendly–certified in Japan (Japan Breast Feeding Association, 2019), such certification might not be the only characteristic that explains the finding.

The reduction in breastfeeding support during the pandemic might be attributable in part to the unique context of Japan. When the study was conducted, no vaccination was available for SARS-CoV-2, and physical distancing was an important means to control disease transmission. Although data collected before the present study showed an infection rate of only 0.02% among pregnant women (Japan Association of Obstetricians and Gynecologists, 2021), medical professionals might have regarded mother–infant separation as a means to reduce the risk of virus transmission. They might also have concluded that the harms of breastfeeding cessation were less than those of COVID-19. In addition, there has been no clear message from the Ministry of Health, Labor and Welfare, or from any other relevant health organization in Japan, that breastfeeding support is an essential health service that should be provided even during the pandemic. Instead, authorities repeatedly recommended that mothers infected with SARS-CoV-2 be separated from their newborns for at least the initial 48 h after delivery (Japan Society for Infectious Diseases in Obstetrics and Gynecology, Japan Society of Obstetrics and Gynecology & Japan Association of Obstetricians and Gynecologists, 2021), which contradicts recent evidence showing that breastfeeding benefits maternal and infant health, regardless of SARS-CoV-2 infection status (Zhu et al., 2021), and that infants should not be separated merely because of the infection (Salvatore et al., 2020; Ronchi et al., 2021; Dumitriu et al., 2021). The absence of an evidence-based recommendation that encouraged breastfeeding support could have reduced awareness among health professionals, especially when the health system was under severe pressure during the pandemic.

Another factor to be considered is that Japan has long failed to develop a clear strategy for breastfeeding promotion that targets health professionals and the public. An unsatisfied intention to breastfeed can lead to maternal guilt, and some people believe that such conflicted feelings are caused by breastfeeding promotion (Matsumoto & Tabuchi, 2019), not by insufficient support. Perhaps in light of this hypothesis, Japanese health authorities have not clearly endorsed exclusive breastfeeding for the first 6 months. The latest professional guidelines for the feeding of infants and young children devotes only two sentences of a 54-page document to summarizing the benefits of breastfeeding (Study Group on Revision of Support Guide for Breastfeeding and Weaning, 2019). The low priority assigned to breastfeeding, even before the pandemic, might have contributed to rapid withdrawal of breastfeeding support to healthy dyads during the pandemic in Japan.

Receipt of a free formula sample or invitation to a free sample campaign was negatively associated with exclusive breastfeeding, besides the incompliance to the Ten Steps. Although Japan voted for the International Code of Marketing of Breast-Milk Substitutes, which prohibits the distribution of free formula samples (World Health Organization, 1981), Japan has no legal regulation on the Code (World Health Organization, 2020). The current study found that most women received the free sample or an invitation to a free infant formula campaign, which was a major risk factor of not exclusively breastfeeding. Aggressive marketing of infant formula is a well-known obstacle to breastfeeding and therefore affects child health (Clark et al., 2020). The results suggest legal regulations on marketing of infant formula in Japan should be considered.

Several limitations of this study should be mentioned. First, the possibility of selection bias cannot be ruled out. The women on the survey company’s panel were not randomly selected from the population of Japan. Further, we did not intend to achieve a high response rate (i.e., we aimed to recruit the first 1,000 respondents among the 4,373 invited women). Mothers with negative experiences of childbirth or breastfeeding might be more likely to ignore invitations to participate in such a survey. Therefore, the current study might have led to the selection of fewer women with negative breastfeeding experiences. Second, all data were collected by self-report, which is subject to social desirability bias. For example, reported breastfeeding outcomes may be better than actual outcomes. Nevertheless, we believe that use of an online panel was appropriate for recruiting postpartum women from a wide geographical area in Japan during a pandemic. Finally, the two groups, women who had a delivery before the pandemic and those who had a delivery during the pandemic, might not be comparative because of the cross-sectional study design. Although the two groups were not significantly different in sociodemographic and obstetric background, the difference in breastfeeding support and outcomes between the two groups can be attributed to unmeasured factors.

Limitations notwithstanding, the present findings highlight how the COVID-19 pandemic might indirectly weaken breastfeeding support for healthy mothers and infants. To our knowledge, this is one of only a few studies to evaluate professional breastfeeding support for healthy dyads during the COVID-19 pandemic. The current study also suggested that breastfeeding support adherence to the WHO recommendations might effectively ensure exclusive breastfeeding during the pandemic. The implication for public health is to emphasize maintaining breastfeeding support for healthy dyads. Considering short-term and long-term breastfeeding benefits (Rollins et al., 2016; Victora et al., 2016), breastfeeding support adherent to the WHO recommendations (World Health Organization, 2017; World Health Organization, UNICEF, 2009) might reduce the indirect adverse effects of the pandemic on non-infected mothers and infants.

Conclusions

In conclusion, the COVID-19 pandemic weakened breastfeeding support for healthy women in Japan; however, support practice that adhered to WHO recommendations appeared to be effective during the pandemic. Because breastfeeding yields long-term health benefits for mothers and infants, we suggest efforts to provide appropriate breastfeeding support even during the pandemic.

Supplemental Information

Supplemental Information 1 Completed STROBE checklist

The STROBE checklist with the page numbers from the manuscript where readers will find each of the items listed below. The page numbers correspond to the submitted manuscript, not the published version.

Click here for additional data file.

Supplemental Information 2 Definitions and measurement of the ten steps to successful breastfeeding

The left column shows the summary of each step. The right column explains how each step was measured or why the step was not measured.

Click here for additional data file.

Supplemental Information 3 Questionnaire with English translation

Click here for additional data file.

The authors would like to thank Dr. Joseph Green for comments and suggestions on an earlier version of the manuscript and David Kipler, ELS, for English editing.

Additional Information and Declarations

Competing Interests

Author Contributions

Human Ethics

Data Availability

The authors declare there are no competing interests.

Keiko Nanishi conceived and designed the experiments, performed the experiments, analyzed the data, prepared figures and/or tables, authored or reviewed drafts of the paper, and approved the final draft.

Sumiyo Okawa, Sarah K. Abe and Takahiro Tabuchi conceived and designed the experiments, performed the experiments, authored or reviewed drafts of the paper, and approved the final draft.

Hiroko Hongo performed the experiments, authored or reviewed drafts of the paper, and approved the final draft.

Akira Shibanuma analyzed the data, authored or reviewed drafts of the paper, and approved the final draft.

The following information was supplied relating to ethical approvals (i.e., approving body and any reference numbers):

This study was approved by the Institutional Review Board of the Osaka International Cancer Institute (No. 20084) and the Ethical Committee of the Graduate School of Medicine, the University of Tokyo (No. 2020336NI).

The following information was supplied regarding data availability:

The data is available at Zenodo: Keiko Nanishi. (2021). Breastfeeding support during COVID-19 pandemic in Japa [Data set]. Zenodo. https://doi.org/10.5281/zenodo.5775279.

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
