# Peer review of "Influence of the COVID-19 pandemic on breastfeeding support for healthy mothers and the association between compliance with WHO recommendations for breastfeeding support and exclusive breastfeeding in Japan"

_PeerJ, doi:10.7717/peerj.13347_

## Round 0.1 · original submission · Major Revisions

Please, revise point-by-point the comments made by the reviewers.

Reviewer 1 ·

Basic reporting

The article is clearly written in professional, unambiguous language. It treats an actual problem: the difficulty to guarantee breastfeeding support during Covid-19 pandemic.

Experimental design

There are some fundamental concerns with the experimental design and with the analysis. The authors should clarify the following sections to avoid confusion and to make this article acceptable for publication.
Material and Methods
Study design:
• Please clarify why you target sample size at 1,000 women.
• I suggest to describe the sampling strategy or cite the journal in which it has been described
• Line 152. Please specify the reference that shows why “the policy of mother–infant separation has not changed as of this writing”.
Infant feeding status:
• According to WHO definition it is not correct to identify the infant feeding status (e.g. "exclusively breastfeeding") upon 24 hours feeding modality.
• Newborns’ age at the moment of the survey "younger than 5 months" is too generic, it needs more explanations. Consider the mean age at the moment of the survey.
I would suggest to detail better the entire paragraph or to remove it from the study as it do not add conclusive data.

Validity of the findings

Analysis
Data on Infant feeding status need to be modified according to previous notes.

Results
Characteristics of participants
Data on Infant feeding status need to be modified according to previous notes.
Line 230 I would suggest to erase “although”
Lines 265-286 detail better the paragraphs or remove it from the study according to previous notes

Discussion
Line 295 modify according to previous notes
Line 317 please you cannot use “exclusive breastfeeding” in this case.

Tab 1 and Tab 2
Modify explanation and Tabs according to previous notes.

Tab 3, Tab 4 and Tab 5
Modify explanation and Tabs according to previous notes or remove them as they are not necessary if you remove data on infant feeding status.

Additional comments

The definition of “Exclusively Breastfeeding” is not used in correct way. This affects all data about “Infant feeding status”. This means the conclusions put forward by this manuscript are not warranted and I cannot approve the manuscript in this form. I suggest to focus attention on breastfeeding support and not on infant feeding status as the data the authors describe do not support any conclusion on infant feeding

Reviewer 2 ·

Basic reporting

In this study, the authors examined maternal breastfeeding intention and professional breastfeeding support before and during the pandemic in Japan and analyzed any differences. They also further assessed if compliance with WHO recommendations for infant feeding support was effective in promoting exclusive breastfeeding during the pandemic. This study is interesting, but it has errors in the interpretation of results, the two groups may not be comparable, and there was very low response rate (22.9%).
English can be improved.
I suggest including a paragraph on the behavior of the COVID-19 pandemic in Japan, i.e., when the first case appeared, what was the incidence and prevalence of COVID-19, were health services affected or not, etc.

Experimental design

The Methods section needs more information
Data collection was started on 15 October 2020 and ended on 25 October 2020. This can lead to bias, as the survey seeks retrospective evidence.
Include the STROBE checklist. Supplemental information.
Lines 124-136. To complement this paragraph I suggest including the STROBE flowchart as Figure 1.
Describe how the survey was validated.
was a pilot test carried out?

Validity of the findings

Results
The interpretation of some results is incorrect. There can be no reduction if the statistical test is not significant.
There was no significant difference in likely to intend to breastfeed during the pandemic (82.2% before vs. 75.6% during the pandemic, p = 0.120).
There was no significant difference en the average number of breastfeeding support steps received was lower during the pandemic (3.24 before vs. 3.01 during the pandemic, p = 0.069), and could not talk with friends about infant care and feeding (10.4% before the pandemic vs. 15.2% during the pandemic; p = 0.169).
The main findings of the study are summarized at lines 278-286. I suggest including this result in the Abstract. In the other cases there are no significant differences.

Discussion
There are two limitations that should be discussed: 1) The main limitation is that the two groups may not be comparable, and 2) very low response rate (22.9%).
Discuss the strengths of the study and the implications for public health.

Additional comments

Abstract. Correct interpretation of results, and include the Conclusion section.

---

## Round 0.2 · accepted · Accept

The comments and points raised by the reviewers have been covered and answered by the authors. The manuscript is now acceptable for its publication.

Reviewer 1 ·

Basic reporting

This second version of the manuscript is correct. The important points of concern have been
clarified and fixed and a positive action can be taken.

Experimental design

No comment

Validity of the findings

No comment

Reviewer 2 ·

Basic reporting

No comments

Experimental design

No comments

Validity of the findings

No comments

Additional comments

No comments